# Bioengineered Organoids Offer New Possibilities for Liver Cancer Studies: A Review of Key Milestones and Challenges

**DOI:** 10.3390/bioengineering11040346

**Published:** 2024-04-01

**Authors:** Abdullah Jabri, Jibran Khan, Bader Taftafa, Mohamed Alsharif, Abdulaziz Mhannayeh, Raja Chinnappan, Alaa Alzhrani, Shadab Kazmi, Mohammad Shabab Mir, Aljohara Waleed Alsaud, Ahmed Yaqinuddin, Abdullah M. Assiri, Khaled AlKattan, Yogesh K. Vashist, Dieter C. Broering, Tanveer Ahmad Mir

**Affiliations:** 1College of Medicine, Alfaisal University, Riyadh 11211, Saudi Arabiarchinnappan@alfaisal.edu (R.C.); aljohara.w.alsaud@gmail.com (A.W.A.); kkattan@alfaisal.edu (K.A.);; 2Tissue/Organ Bioengineering and BioMEMS Lab, Organ Transplant Centre of Excellence (TR&I Dpt), King Faisal Specialist Hospital and Research Centre, Riyadh 11211, Saudi Arabia; 3Department of Medical Laboratory Technology, Faculty of Applied Medical Sciences, King Abdulaziz University, Jeddah 21423, Saudi Arabia; 4Pathology and laboratory Medicine, Perelman School of Medicine, University of Pennsylvania, Philadelphia, PA 19104, USA; 5School of Pharmacy, Desh Bhagat University, Mandi Gobindgarh 147301, Punjab, India; shababmir123@gmail.com

**Keywords:** liver, cancer, biopsy, stem cells, organoids, 3D culture, tissue engineering, regenerative medicine

## Abstract

Hepatic cancer is widely regarded as the leading cause of cancer-related mortality worldwide. Despite recent advances in treatment options, the prognosis of liver cancer remains poor. Therefore, there is an urgent need to develop more representative in vitro models of liver cancer for pathophysiology and drug screening studies. Fortunately, an exciting new development for generating liver models in recent years has been the advent of organoid technology. Organoid models hold huge potential as an in vitro research tool because they can recapitulate the spatial architecture of primary liver cancers and maintain the molecular and functional variations of the native tissue counterparts during long-term culture in vitro. This review provides a comprehensive overview and discussion of the establishment and application of liver organoid models in vitro. Bioengineering strategies used to construct organoid models are also discussed. In addition, the clinical potential and other relevant applications of liver organoid models in different functional states are explored. In the end, this review discusses current limitations and future prospects to encourage further development.

## 1. Introduction

According to the latest World Cancer Statistics, liver cancer ranks as the third leading cause of cancer-related death among all malignancies globally, posing a serious threat to human health [1]. Hepatic cancer causes a significant economic burden affecting tens of millions of families worldwide. Primary liver cancer includes different pathologic categories. Hepatocellular carcinoma (HCC) and intrahepatic cholangiocarcinoma (iCCA) are the two main histologic types of primary liver cancer, which differ in their epidemiology and etiology [2]. Hepatocellular carcinoma comprises more than 90% of the primary cancer of the liver. The main risk factors for the initiation and progression of hepatocellular carcinoma (HCC) are infections caused by hepatitis B or hepatitis C viruses. In areas of high hepatitis B endemicity, the hepatitis B virus is transmitted in large numbers from mothers to infants during childbirth, and transmission of hepatitis B and hepatitis C viruses generally occurs through the usage of unsafe needles and injection-related medical procedures. Hepatocellular carcinoma can also be caused by several other factors, including alcohol consumption, non-alcoholic liver steatohepatitis (NASH), non-alcoholic fatty liver disease (NAFLD), obesity, diabetes, and aflatoxin intake.

Intrahepatic cholangiocarcinoma (iCCA), on the other hand, originates in the bile ducts within the liver lobules [3]. Although risk factors worldwide are unclear, the best-known preventable causes of hepatobiliary cancer are attributed to the foodborne trematode parasites (Opisthorchis viverini and Clonorchis sinensis), which are found in certain endemic regions [4]. Other known risk factors responsible for developing and progressing iCCA include primary sclerosing cholangitis, Caroli’s disease, and hepatolithiasis [4]. Lesser-known subtypes of liver cancer include hepatoblastoma (a rare pediatric cancer) and angiosarcoma, which is associated with long-term occupational exposure to vinyl chloride monomer, among other risk factors [5]. Although each of these tumors has distinct genetic profiles and characteristics, they all have an extremely unfavorable prognosis and are marked by a high likelihood of recurrence after surgical removal and low survival rates in case of metastasis [6]. Historically, finding effective treatments for patients with liver cancers, particularly hepatobiliary cancers, has been challenging due to the lack of appropriate research models. Hence, accurate estimation of the pathophysiology and treatment of hepatic cancer by subtype requires the development of new therapeutic and diagnostic substitutes to address the existing challenges in hepatology research [7,8,9,10].

Currently, preclinical models of liver research primarily rely on two-dimensional cell cultures of mammalian cells and animal-based experimental strategies [11]. The former strategy involves attaching one type of primary or immortalized cell line to a plastic substrate, either in the form of a monolayer of cell sheets in a culture flask or in a flat Petri dish. Conventional two-dimensional cultures are generally characterized by simplistic two-dimensional interactions of cells and exhibit homogeneity when exposed to exogenous substances or drug molecules. Adherent 2D culture methods are still dominant in many biological studies, but they are far away from the native physiological or pathophysiological conditions, leading to erroneous results and enormous economic costs. The simplistic approach of 2D substrates makes it difficult to mimic the comprehensive liver tissue microenvironment of the developing cancer. To overcome this challenge, preclinical animal models have been used to mimic and investigate the in vivo human body’s microenvironment [12].

However, animal models are often limited by intra- and interspecies variations and human physiological differences. In addition, they are very costly to breed, house, and maintain. Thus, in vitro 3D models that recapitulate the physiological similarities of native tissues are a promising alternative to bridge the gap between 2D culture system and animal-based models [13,14,15,16,17,18,19].

### Current Methodologies for the Detection and Treatment of Hepatocellular Carcinoma

The long-term survival of patients following liver cancer treatment is often impacted by high recurrence rates, which could reach up to 40–70% within five years [20,21]. Early detection and identification of recurrence are crucial for optimal management of HCC in the long term. Radiological imaging is a commonly used, non-invasive method for evaluating treatment response, especially post-resection or locoregional therapy. The Liver Imaging Reporting and Data System (LI-RADS) and the European Association for the Study of the Liver (EASL) guidelines include ultrasound, especially contrast-enhanced ultrasound (CEUS), along with contrast-enhanced CT (CECT), Contrast-Enhanced Magnetic Resonance Imaging (CE-MRI), and Positron Emission Tomography (PET) scans, as the recommended modalities [22]. CEUS, in particular, provides real-time imaging of blood flow in lesions, enabling accurate differentiation between viable and necrotic tumor tissue and improving the precision and accuracy of assessing treatment efficacies [23]. Nonetheless, patient factors, such as cirrhosis, extensive fibro-fatty changes, body type, intestinal gas, and deep lesions, may make it difficult to identify recurrence using ultrasound quickly [24,25]. CECT and CE-MRI encounter fewer limitations in this regard. Moreover, they offer the advantage of unraveling more intricate anatomical insights, and CE-MRI is better at demonstrating exceptional differentiation of soft tissues. The customary practice for assessing treatment response involves the combined utilization of CT and MRI [22]. Functional imaging techniques, such as Positron Emission Tomography (PET) and Single-Photon Emission Computed Tomography (SPECT), provide valuable insights by detecting metabolic changes within lesions [26]. The published literature has also demonstrated that a PET scan combined with CT or MRI can provide higher detection efficacy [27].

Until now, numerous biomarkers have been identified and proposed for diagnosing hepatocellular carcinoma and evaluating therapeutic efficacy [28]. In recent years, the utility of biomarkers has become increasingly important for determining a drug’s mechanism of action, investigating toxicity and efficacy signals, and estimating patient response to systemic therapy over locoregional or surgical therapies [29]. The most recognized and utilized biomarker in the literature for HCC diagnosis and management is alpha-fetoprotein (AFP). Kim and co-workers evaluated AFP and radiographic changes in 108 patients with hepatocellular carcinoma and concluded that AFP changes provide prognostic information after management with immune checkpoint inhibitors [30]. He and his team performed a meta-analysis of twenty-nine studies on prognosis prediction with AFP [31]. The authors discovered that overall survival was significantly associated with AFP responsiveness to post-treatment. However, they also highlighted the unreliability of AFP in certain cases. This is mainly due to the fact that certain hepatocellular carcinomas are AFP-negative and that AFP might be released in liver diseases, such as cirrhosis [32,33]. In a prospective study in which Zhang and co-workers investigated 1338 HCC patients, the seropositivity rate of AFP was 46%. Therefore, they proposed the combination of AFP with human cervical cancer proto-oncogene 1 (HCCR-1), a biomarker for which 51.3% of their patients were seropositive. Indeed, the combination of AFP with various other biomarkers, such as Glypican-3 (GPC-3) and prothrombin induced by vitamin K deficiency or antagonist-II (PIVKA-II), or with other diagnostic indicators, such as neutrophil-to-lymphocyte ratio, has been implemented to improve diagnosis rates [28,29,30]. The clinical utility of biomarkers is still under discussion, largely due to their low sensitivity and specificity to HCC. Additionally, authors differ significantly regarding the cut-off points for assessing tumor progression with these biomarkers [29,30,31,32,33,34,35,36].

Over the past two decades, the treatment landscape for hepatocellular carcinoma has evolved dramatically, and now there are multiple options available depending on tumor stage and liver function. Nevertheless, the difference between surgically resectable and advanced disease poses a significant challenge in medical decision making, prompting the use of various staging classification systems. The Barcelona Clinic Liver Cancer (BCLC) algorithm-based classification system, generated using comprehensive data from various cohort studies and randomized clinical trials (RCTs), is the most universally accepted cancer staging system [37,38]. The BCLC system categorizes HCC patients into five stages (0, A, B, C, and D) and predicts and assigns therapy recommendations (either curative or palliative) based on three main prognostic variables: clinical status of tumor (tumor size, number of nodules, extrahepatic spread, and portal invasion), the function of the liver (assessed using the Child–Pugh score), and the status of the tumoral and cirrhotic factors (based on Eastern Cooperative Oncology Group performance status) [39].

HCC patients in early stages (stage 0-A) are generally recommended for curative treatment (resection, ablation, liver transplantation), patients in the intermediate stage (stage B) are treated with trans-arterial embolization (TAE), and patients in the advanced stage (stage C) are treated systemically. Patients with a very poor prognosis or life expectancy are generally classified as ineligible for any treatment (stage D) [40,41,42,43]. Existing treatment options for patients with advanced-stage hepatocellular carcinoma remain seriously dissatisfying due to limited response sensitivities to different chemotherapy agents. Resistance to current systemic therapies is largely associated with tumor cell plasticity or “hide-and-seek” behavior of cancer cells, epithelial–mesenchymal transition and transdifferentiation, existence of cancer stem cells, and immune-excluded phenotype [44,45,46,47,48]. In addition, the persistence of a tumor-promoting environment within the fibrotic zones promotes cancer cell crosstalk, which influences extracellular matrix deposition and ultimately increases the risk of recurrence patterns after curative treatment. Collectively, there is a growing and unmet medical demand for developing novel technologies for the early diagnosis and treatment of hepatocellular carcinoma.

In this regard, ex vivo organoid models using surgically resected patient tumors are considered promising alternatives to address existing challenges, such as studying patient-specific developmental stages of HCC, selecting appropriate therapy (chemotherapy or radiotherapy), therapy sensitivity screening, and improving treatment efficiency [49,50,51].

## 2. A Brief History of Organoids

The development of organoid-like structures dates back to 1907 when Wilson HV first demonstrated how whole sponges may be artificially reared through self-organization of dissociated sponge cells in vitro [52]. Over the past decades, several research groups have applied dissociation–reaggregation experimental strategies to develop miniaturized model systems of different bio-constructs (the reader is referred to more specialized reports (Figure 1) [52,53,54,55,56,57,58,59,60,61,62,63,64,65,66,67,68,69,70,71]). A real shift in the modern field of organoid biology started in 2009 when Sato et al. reported that adult intestinal stem cells expressing G protein-coupled receptor 5 (Lgr5), which contains leucine-rich repeats, generate intestinal organoids in a 3D microenvironment [72]. The authors reported that murine intestine-derived adult stem cells self-organize and differentiate into crypt-villus in Matrigel (a substrate for 3D cell culture). The results of this benchmark study paved the way for further developments in organoid research using cells derived from different source organs. The technique was subsequently refined to generate human intestinal organoids and organoids from various organs in which leucine-rich repeat-containing G protein-coupled receptor 5 (Lgr5+) progenitor cells are present, including the colon, stomach, and liver [63]. The isolation of individual Lgr5+ cells from the livers of adult mice and their cultivation into hepatic organoids unveiled their colony-forming capacity [66]. Extended cultivation confirmed their ability to self-renew indefinitely [72].

In 2013, Takebe and co-workers successfully produced liver buds and liver organoids in vitro by co-culturing human-induced pluripotent stem-cell-derived liver endodermal cells with stromal cells [66]. After forming liver-bud-like structures, the authors injected the engineered organoids into immunodeficient mice. Interestingly, the vessels of the artificial constructs invaded the host vascular network, and the liver bud mimics induced in vitro were nearly identical to their in vivo counterparts. Following transplantation, bioartificial liver buds facilitated the development of functional liver tissue in mice models, offering a promising avenue for regenerative treatments in cases of organ failure. This process faithfully recapitulated the cellular transformations observed during embryonic organ bud development [73]. Subsequently, the same research team developed a large-scale organoid production platform capable of efficiently producing homogeneous small liver bud constructs [74,75]. Hence, a broad spectrum of organoid models derived from both adult primary cells and pluripotent stem cells has emerged, underscoring the remarkable versatility of this technology [76,77,78]. In the wake of these advancements, liver organoids have become invaluable tools in research, facilitating investigations into liver development, diseases, and drug toxicity [79]. Their application has enabled the study of the mechanisms underpinning liver disorders, including liver cancer, viral hepatitis, and non-alcoholic fatty liver disease [80]. Continued progress in organoid technology holds the potential to revolutionize regenerative medicine, with the prospect of developing transplantable organs or seeding bio-artificial liver devices akin to kidney dialysis machines.

**Figure 1 bioengineering-11-00346-f001:**
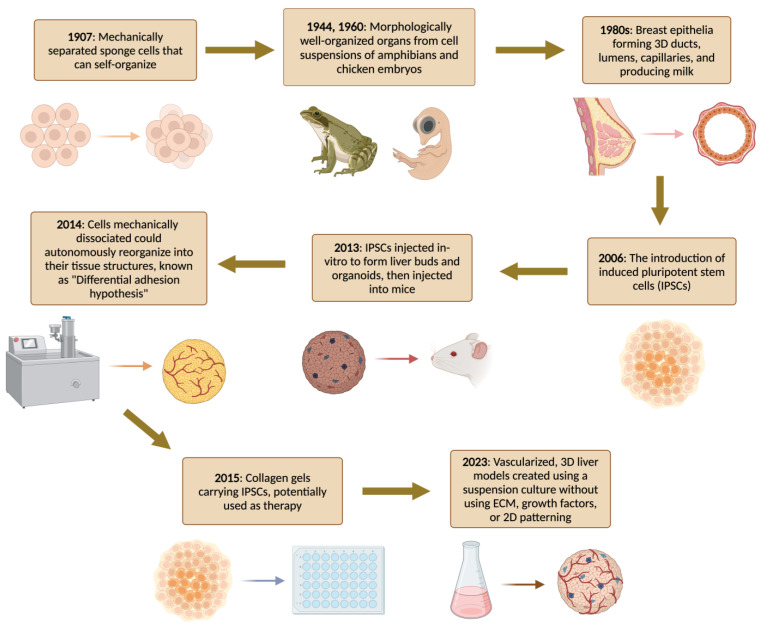
Timeline of the history of the generation of organoid cultures [52,53,54,55,56,57,58,59,60,61,62,63,64,65,66,67,68,69,70,71].

### Overview of the Organoids Concept

The establishment of liver organoids is based on fundamental biological principles of organogenesis and maturation, in which stem cells differentiate and self-assemble into physiologically active immature and mature tissues and organ structures regulated by various signaling molecules. Mimicking these innate features to develop and optimize stem-cell-derived organoid models requires exogenous cellular and tissue microenvironment stimuli. The cell morphogenesis, physiology, fate determination, and organization that drive organogenesis are directed by the combined use of multiple biochemical signaling factors and biophysical restraints from the surrounding extracellular microenvironment and adjacent cells [81,82].

For example, during the early stages of fertilization and embryogenesis, the embryo reorganizes and immediately divides into different cell types in a constrained neighborhood until an appropriate embryonic cavity is formed. At this stage, the fertilized egg divides into multiple cells with slight changes. Consequently, the cells become more cohesive and compact until blastocyst formation occurs. In addition, cell polarization is facilitated by paracrine signaling pathways and mechanical cues exerted by neighboring cells, which ultimately allow for the formation of germ layers and the organization of tissue structures [83,84]. Researchers in the field of tissue engineering and organoids-based regenerative medicine have succeeded to some extent in mimicking these basic principles of embryology to establish organoid constructs simply by controlling cell assembly and maintaining shape, size, and coordination inputs. This review will shed light on the cell and tissue engineering principles used to create organoid systems and their application in liver research.

## 3. Cell Source and Biological Factors for Growing Liver Organoids

Adopting stem cells for organoid growth is one of the most critical advances in tissue engineering and regenerative medicine research. Organoids can be established from pluripotent stem cells (PSCs) or organ-specific adult stem cells (ASCs). Liver progenitors expressing the membrane receptor Leucine-rich repeat-containing G protein-coupled receptor 5 (LGR5) are widely used for generating liver organoids. It has been well documented that LGR5-expressing cells possess the potential for self-renewal and self-assembly and that they can generate organoids of functional cholangiocytes and hepatocytes (Figure 2) [85,86,87].

Published studies have indicated that liver-derived LGR5+ progenitors could be stimulated to generate mini liver tissue or organoids while retaining their “non-stem-cell” characteristics after a long-term 3D culture microenvironment. PSCs can self-renew and differentiate into specific lineages or any cell type in the body, including liver cells. Protocols have been developed for the gradual differentiation of PSCs into hepatic cells, producing liver organoids of hepatocytes or cholangiocytes that have liver-like functions, such as albumin, urea, and bile acid synthesis [66,88]. This approach has proven to be highly effective with both human embryonic stem cells and human-induced pluripotent stem cells, and it has been recently employed for the in vitro modeling of liver diseases [89]. However, the complexity of differentiating human pluripotent stem-cell-derived hepatic cells and the potential formation of teratomas due to any remaining undifferentiated cells in the system present significant challenges for the derivation and application of PSC-derived hepatic cells in cell and organoid-based therapies [90].

In addition, when hepatocytes are cultured in traditional monolayers, they typically undergo dedifferentiation, leading to a loss of phenotype and function [91]. This lack of morphology and function often results in the inability to detect metabolism-mediated hepatotoxicity of drugs in vitro [92]. Nevertheless, the differentiation of pluripotent stem cells, such as embryonic stem cells (ESCs) and induced pluripotent stem cells (iPSCs), into hepatic cells has been a key strategy since organoids’ early development. This method allows for the gradual differentiation of stem cells into liver organoids, mimicking the developmental process observed in natural liver formation [93].

Growth factors play a crucial role in the in vitro growth of organoids by manipulating the regulatory activities of cells related to self-renewal, expansion in a 3D microenvironment, differentiation after exposure to a specific culture medium, and self-organization into stable or static structures. During embryonic development, early liver development begins with the patterning of the endodermal layer, which also gives rise to the endodermal tissues of the pancreas, intestinal tract, and gut tube [94]. Researchers have therefore mimicked the natural phenomenon by using culture media supplemented with several growth factors and biomolecules that play key roles in signaling pathways crucial for proliferation, differentiation, maturation, and migration in the endodermal system. The major biological factors widely used for the growth, maintenance, and genetic manipulation of organoids (including liver) include noggin, R-spongin 1 (Wnt pathway enhancer), epidermal growth factor (EGF), hepatocyte growth factor (HGF), fibroblast growth factor (FGF), transforming growth factor-β (TGF-β), TGF-β inhibitor (A83-01), cAMP pathway agonist forskolin (FSK), and retinoic acid (RA) [95,96,97].

## 4. Bioengineering Strategies for Growing Liver Organoids

A variety of tissue engineering techniques have been investigated to support cells’ growth, proliferation, and differentiation into organoids of hepatocytes and cholangiocytes. Three-dimensional cultures based on hydrogels or scaffold matrices prepared from biocompatible materials of natural or semi-synthetic origin provide a biomimetic in a three-dimensional microenvironment for cells to differentiate and organize into functional liver organoids over time [98,99]. To effectively model liver functionality in vitro, it is critical not only to create an inbuilt vasculature, but also to tailor the cellular phenotype by modulating structural properties of the matrix with appropriate stiffness and porosity. The hydrous microenvironment and related properties of the extracellular matrix-mimicking culture system are essential for ensuring a constant and efficient permeation of nutrients and oxygen supply, which plays a vital role in supporting the functional integrity of the liver model [100,101]. Natural biomaterials, such as hydrogels, present several advantages, including their capacity to mimic the extracellular matrix (ECM) and sustain a supportive microenvironment, offering mechanical and biochemical signals to cells [79,102,103,104].

Currently, the establishment of 3D hepatic organoids predominantly relies on Matrigel, a natural hydrogel derived from a protein mixture secreted by Engelbreth–Holm–Swarm murine sarcoma cells [105]. Matrigel serves as a supportive substrate that mimics the complex extracellular matrix (ECM) environment of the liver [106]. To guarantee uniformity and repeatability, it is crucial to establish well-defined culture conditions for organoids, especially with regard to their composition, to facilitate future clinical applications [107]. Over the past decade, there has been a notable progression in organoid culture techniques. Initially reliant solely on Matrigel, these techniques have since advanced to incorporate biohybrids and semisynthetic or synthetic hydrogels, offering greater control and standardization in the generation of organoids [108]. While Matrigel provides valuable support for cells and facilitates their physical attachment and self-organization, which is crucial for the development of organoids, it comes with several limitations. These include inconsistencies between batches and variations across species, as well as concerns regarding the transmission of animal pathogens and potential incompatibility or immunogenicity issues in clinical applications due to having unidentified components from its animal origin [100,109]. The development of liver organoid models using chemically defined synthetic polymers eliminates the need for animal components. It also aims to overcome the batch-to-batch variability associated with traditional matrices like EHS and produce highly reproducible organoids, making them more suitable for disease modeling and clinical applications [110,111]. Liver organoids have been successfully generated within the PEG-RGD, (PIC), and laminin-111 hydrogel, and the stiffness of the hydrogel has been manipulated to replicate fibrotic liver disease. Another study demonstrated the generation of human liver organoids using polyisocyanopeptides (PIC) and laminin-111 [111]. Additionally, natural polymer hydrogels, such as fibrin/laminin, have been employed to generate liver organoids due to their ability to provide appropriate biochemical and physical support for organoid formation and expansion [110]. Hence, incorporating biomaterials, whether synthetic or derived from natural sources, has significantly accelerated the advancement of liver organoid development.

### Microfabrication Strategies to Control the Assembly of Organoids

The microfluidic-based approach employs microfluidic devices that enable the growth and differentiation of liver cells. These devices provide precise control over the cellular microenvironment, including the composition of the media and the flow of nutrients and waste. Microfluidics refers to a technology that enables the manipulation of small amounts of fluids within tiny channels. This innovative approach has found application in creating microenvironments that closely resemble the natural conditions of organs [112]. This can exert precise control over the microenvironment, enabling the establishment of nutrient and oxygen gradients, as well as facilitating cell–cell interactions [112,113,114]. The aim is to effectively provide nutrients and oxygen to the inner components of the organoids [100]. In their study, Rennert et al. successfully developed a functional 3D human liver model within a microfluidic biochip. The liver organoids in this model receive a continuous supply of nutrients and oxygen through a fluid flow system, which closely mimics the structure of the liver sinusoid. To measure oxygen consumption, a luminescence-based sensor was integrated into the microfluidic chip. The results confirmed that the introduction of microfluidic flow using a perfusion system led to an enhanced expression of liver transporter proteins and promoted the formation of hepatocyte microvilli [114]. Another microfluidic chip was developed by Banaeiyan et al. to culture HepG2 cells and hepatic liver cells, aiming to replicate the liver lobule microenvironment. The chip featured a tissue-like hexagonal design and incorporated micro-channels to simulate the convection–diffusion mechanism of blood circulation [115]. Prodanov and co-workers developed a microfluidic chip with two chambers separated by a porous membrane. They aimed to recreate the liver sinusoid using human cells for 28 days and successfully maintained a 3D model [116]. The application of an efficient digital microfluidic system has been demonstrated for drug screening and evaluation of hepatotoxicity [117]. Esch and his team combined the Liver Chip and Intestine Chip to evaluate toxicity in the digestive system. The liver compartment contained HepG2/C3A cells (which are hepatocellular carcinoma cells) cultured in a silicon chip, while the intestine compartment included a co-culture of Caco-2 (Colon carcinoma) and HT29-MTX (mucus-secreting colon epithelium) cells. Interestingly, linking the intestine compartment to the liver compartment via microfluidic channels exacerbated this effect, resembling the first-pass metabolism despite reducing nanoparticle exposure in the Liver Chip [118]. The utilization of microfluidics in organoid development offers several benefits, including precise control over the microenvironment, efficient mass transport facilitated by fluid flow, and the ability to integrate with different sensors and actuators. However, there are also limitations to consider, such as the challenges of standardization and scalability. Additionally, the operation of microfluidic systems may require external pumps, tubing, connectors, and valves [93]. The approaches mentioned above have emerged as the most widely used and have demonstrated their effectiveness in generating functional liver organoids. They have the potential to revolutionize disease modeling, drug testing, and personalized medicine in the future.

## 5. Organoids for Liver Cancer Research

Liver cancer organoids may be derived from adult tissue biopsies or by reprogramming patient-derived cells with growth factor cocktails (Figure 3 and Figure 4). Using patient-derived organoid models to investigate mechanisms of chemotherapy resistance and to reverse drug resistance by targeting cancer stem cells and oncogenes is one of the hottest topics in cancer research. Over the years, the tumor microenvironment (TME) has been widely studied and targeted by various drugs, such as PD-L1 inhibitors and anti-angiogenics [119]. Yet, altered microenvironmental elements throughout therapy result in acquired drug resistance, leading to decreased clinical effectiveness [119]. Observing these changes in the TME could aid in developing new drugs for advanced HCC. Organoids have provided an excellent 3D environment in preclinical studies for studying TME resistance and potential new targets. Cancer-associated fibroblasts (CAFs), a major component of the TME, are known to be involved in the progression of cirrhosis and are described as promoting tumor proliferation and invasion. CAFs have been co-cultured with primary liver organoids and revealed to promote resistance to chemotherapeutic drugs, including Sorafenib, Regorafenib, and 5-fluorouracil [73]. Sorafenib is an anti-cancer agent that targets various kinases involved in certain oncogenic pathways [120]. The role of CD 44 positive HCC cells and certain hedgehog signaling pathways in Sorafenib has been described [121,122]. Co-administration of hedgehog signaling pathways with Sorafenib exhibited increased sensitivity to Sorafenib [123]. Martin and co-workers discussed the resistance to trans-arterial chemoembolization (TACE) of HCC, noting that up to 40% of patients do not respond to treatment [124]. PKM2, a crucial enzyme in glucose metabolism, was identified as participating in cancer cell metabolism.

**Figure 3 bioengineering-11-00346-f003:**
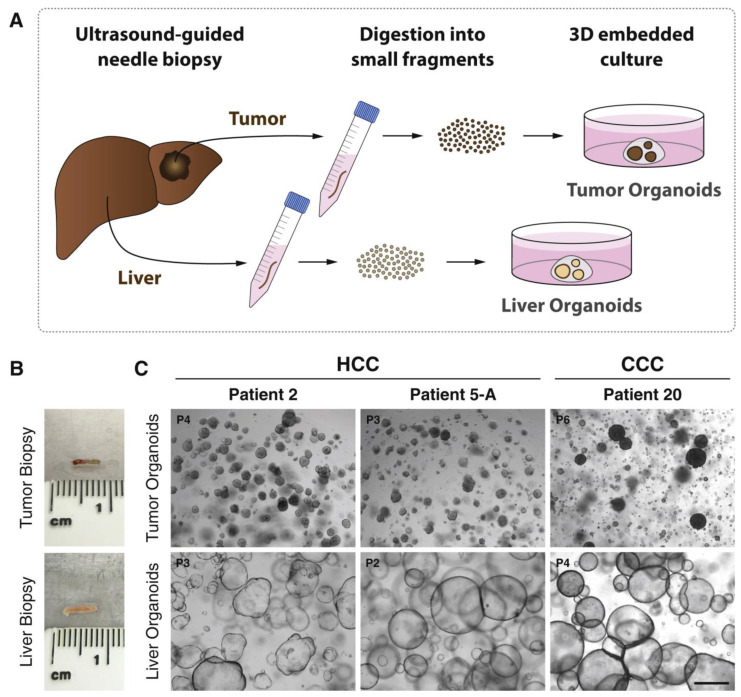
Generation of cancer organoid from human liver biopsies of hepatocellular carcinoma and paired non-tumor liver tissues. (**A**) Schematic illustration; (**B**) image of biopsy pieces employed for growing organoids; (**C**) images of a tumor and paired non-tumor liver tissue organoids captured using a bright-field microscope (Scale bar: 500 μm). Abbreviations: hepatocellular carcinoma (HCC); cholangiocellular carcinoma (CCC); passage (P). Figure 3 is adopted with copyright permission from [125], Elsevier.

Nevertheless, numerous targets for novel anti-cancer agents have been established via organoid research. These would permit the establishment of precision therapy for HCC patients, thus reducing overall side effects in contrast with traditional chemotherapy. Inhibitors of purine metabolism are commonly used to treat various cancer types. However, the significance of purine metabolism in liver carcinogenesis remains uncertain. Chong and co-workers discovered the significance of amplified de novo purine synthesis in HCC using cell lines and organoids. E2F1, a transcriptional factor, was observed to regulate the PI3K pathway, which promotes carcinogenesis by upregulating purine synthesis [126]. Purine metabolism, therefore, could be a potential target for personalized therapy. Preferential deposition of fluorescent nano-diamond particles (FDP) coated with doxorubicin, a chemotherapy agent, allows growth inhibition and cell death selectively in tumor cells, thus lowering adverse effects [127]. Oroyxlin A (transketolase inhibitor) is a novel anti-cancer drug studied using liver organoids. Its impact on inhibiting transketolase, a rate-limiting enzyme in de novo nucleotide synthesis, suppressed growth and cell death [128]. Lim and coworkers employed liver organoids to observe and deduce rational drug combinations for proteosome inhibitors, thus overcoming the limited furtherance in combinational therapy, along with discovering enhanced efficacy and clinical outcomes [129]. Omacetaxine is an FDA-approved global protein synthesis inhibitor used as an anti-cancer agent for chronic myelogenous leukemia [130]. Its effect on six hepatocellular carcinoma patient-derived cell lines was assessed, which revealed early apoptosis, late apoptosis, or both in all six lines [131].

**Figure 4 bioengineering-11-00346-f004:**
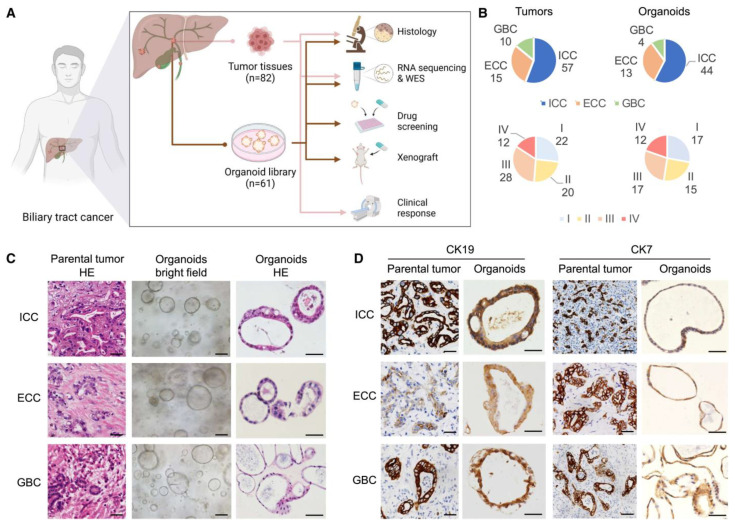
Establishment of patient-derived biliary tract cancer organoids. (**A**) Experimental strategy for biopsy collection from biliary tract of cancer patients. (**B**) Pie charts comparing stratification of all original tumors versus stratification of all tumor-derived organoids based on histology (**top**), tumor-node-metastasis (TNM), and disease stage (**bottom**). (**C**) Microscopy images for histology staining (H&E) and bright-field imaging of patient-derived organoids and matched primary tumors from three main histological subtypes (scale bars, 50 μm). (**D**) Immunohistochemistry analysis of biliary tract cancer markers (CK19 and CK7) in original tumor samples and BTC organoids (scale bars, 50 μm). Abbreviations: biliary tract cancer (BTC); intrahepatic cholangiocarcinoma (ICC); extrahepatic cholangiocarcinoma (ECC); and gallbladder carcinoma (GBC). Figure 4 is adopted with copyright permission from [132], Elsevier.

Liver organoids have been pivotal in exploring the connection between hepatocytes and cholangiocarcinoma (CC) in aiming to unveil CC’s etiology [72]. In an interesting study, Sun et al. utilized human-induced hepatocytes to generate liver organoids and overexpress ten genes enriched in intrahepatic cholangiocarcinoma (ICC). This led to notable changes, including the loss of circular morphology, mucous vacuole development, increased nucleus-to-cytoplasm ratio, nuclear atypia, duct or cavity structures, mucin release, and upregulated NOTCH signaling genes. After being orthotopically implanted in mice, these organoids resulted in a 100% tumor development rate, resembling ICCs due to specific ICC markers (CK19, SOX9). This suggested an important role of the RASG12V gene in hepatocyte-to-ICC conversion [133]. Studies using liver cancer organoids from PPTR mice (Prom1 overexpression, Pten, and TP53 loss) demonstrated strong stem cell properties, potentially leading to invasive and metastatic tumors [134]. Interestingly, some organoids showed features of both hepatocellular carcinoma and cholangiocarcinoma, which can differentiate into mature liver cells, reducing intrahepatic cholangiocarcinoma aggressiveness by blocking the Wnt signaling pathway [135]. These findings highlight the potential for gene or pathway targeting to prevent cholangiocarcinoma. Organoids have been important in investigating mitochondrial function and energy metabolism in cholangiocarcinoma. The Warburg effect underscores glycolysis as a primary energy production mechanism in tumor cells, making mitochondria crucial [136]. Studies found altered mitochondrial shape and reduced metabolism in cholangiocarcinoma organoids with knocked-out mitochondrial fusion genes (OPA1 and MFN1), leading to decreased oxygen consumption and ATP production. Inhibiting apoptosis prevented cholangiocarcinoma organoid development [137]. Another study cultured cholangiocarcinoma organoids without glucose, observing reduced proliferation, altered shape, increased stem cell markers, enhanced AKT phosphorylation, and reduced gemcitabine sensitivity [138]. Inhibition of AKT phosphorylation reversed stem-cell-like features and gemcitabine resistance. This suggests that cholangiocarcinoma cells can adapt to glucose shortage by enhancing stem cell characteristics through AKT phosphorylation.

Hepatitis infects more than 354 million people worldwide and amplifies the risk of HCC [WHO]. A characteristic of the hepatitis B virus (HBV) is the formation of a stable, persistent cDNA episome. This incorporates into the host DNA, rendering it resistant to antiviral drugs [139]. Due to the dysregulation of significant host factors and antiviral targets in 2D culture systems, 3D organoid systems allow for improved research in studying virus–host interactions as well as potential therapeutics [140]. Liver organoids derived from adult tissue have been infected with HBV to study pro-apoptotic drugs that promote cell death via the extrinsic pathway to eliminate hepatocytes with HBV DNA [141]. Hepatitis C virus (HCV) is another less commonly infecting virus leading to liver cirrhosis and HCC. Hepatocytes in vivo exhibit polarity, allowing the HCV virus to enter via certain tight junctions [142] Huh-7.5 hepatoma cells, utilized for conducting in vitro studies, express poorly in 2D but express excellent polarity in 3D organoids, which have allowed for ameliorated understanding of the HCV life cycle [142]. Improved binding of HCV has been observed in Huh-7 3D models, showing polarity over 2D models possessing poor polarity [143].

### Current Limitations of Organoid Models

Organoid technology is still constrained by several hurdles and limitations. The fundamental obstacle associated with this technology is that the culture, maintenance, and expansion of organoids are heavily dependent on Matrigel, a 3D culture matrix extract derived from Engelbreth–Holm–Swarm mouse sarcoma cells. While Matrigel serves as a commercially accessible hydrogel substrate, these extracts encounter issues linked to inconsistencies in composition and reproducibility across different batches. Furthermore, these extracts may harbor unknown pathogens and are highly immunogenic, making them unsuitable for transplantation into human patients and severely hampering the clinical application of organoids. A major drawback of adult stem-cell-derived liver organoids is that they are derived from epithelial cells and utilize a simplified extracellular matrix (ECM) that lacks structural and functional compartmentalization [76,144]. Existing organoid culture systems are monocultures and lack innervation, vascularization, and immune interactions, making them inadequate for understanding the full spectrum of hepatocarcinoma. Therefore, there is a strong need to use approaches, such as tailored matrices and co-cultures, to generate organoids with specific cell populations, such as mesenchymal cells, endothelial cells, and immune cells, in addition to hepatocytes [125,132]

Another drawback is the limited effectiveness of organoid models. A study by Nuciforo et al. on organoid models of hepatocellular carcinoma showed that it is difficult to culture hepatocellular carcinoma cells [125]. The success rates for developing hepatocellular carcinoma organoids (HCCOs) from human primary hepatocellular carcinoma have been reported as ~26%. However, for other cancers, the success rate is significantly higher. Success rates for colorectal cancer are approximately 90% [145], and for pancreatic cancer, they are around 75–83% [146]. The variability among organs indicates that the success rate for hepatocellular carcinoma is much lower than for other cancers. A possible reason for the low success rate could be that the generation of hepatocellular carcinoma organoids is limited to a subset of HCCs. Furthermore, only poorly differentiated tumors are able to produce HCC organoids, which, according to Nuciforo et al., is strongly associated with the histopathological features of the tumors [125]. This could be because the highly differentiated, slowly growing tumors do not reach the threshold of cell proliferation rate required for the development of HCC organoids. Similarly, according to Broutier and coworkers, there is a significant correlation between the original tumor’s proliferation index and success rate. In samples taken from tumors with proliferating cells greater than 5%, the success rate of establishing organoid cultures was 100% [147]. Future studies may seek to culture HCCOs from lower-grade tumors to capture the entire disease spectrum.

Developing liver organoid cultures is often costly and requires skilled professionals, specialized techniques, and equipment. For example, laboratories often produce conditioned media to save experimental costs. Unfortunately, this leads to issues of batch-to-batch variability and, in some instances, small amounts of serum (0.5–1%) are present in the final organoid growth medium, jeopardizing the consistency of the organoid model. A typical base medium for mammalian cell culture is advanced DMEM/F12. However, DMEM/F12 cannot be used for the culture of liver organoids prior to the addition of multiple agents [148]. To encourage the advancement of organoid technology and optimize its application in biological research, the availability of appropriately designed and reasonably priced media is essential. Variation among different iPSC lines and their derivatives continues to be a major challenge, especially for iPSC-derived organoids, when iPSCs and their derivatives are used for disease modeling and cell therapy. When generated from different individuals or iPSC core facilities, variation between and within iPSC lines is frequently observed in distinguishing iPSC tumorigenicity, genomic instability, epigenetic status, and maturation status. Successful development of “comparable” iPSCs and their derivatives requires quality characteristics that produce consistent, high-quality iPSCs.

Therefore, in 2018, the Global Alliance for iPSC Therapies in the UK identified the creation of QC guidelines for clinical-grade iPSC production [149]. Identity verification, microbiological sterility, endotoxin, genetic fidelity and stability (karyotyping and residual vector testing), potency assessment, expression of pluripotency markers, and post-thawed viability are among the essential quality characteristics for clinical-grade iPSC creation. However, there may still be variations during the purification process, iPSC differentiation, iPSC expansion, reprogramming, colony selection, culture system selection, and iPSC reprogramming within various iPSC cell banks. Such issues must be resolved through routine and ongoing validation of the iPSCs. Moreover, recent studies on genetic and epigenetic differences in iPSCs have highlighted iPSC safety-related concerns.

Genome instability, single nucleotide variants, choroidal neovascularization (CNV), and loss of heterozygosity are examples of genetic changes found in iPSCs. Through reprogramming and prolonged in vitro culture, these mutations can be introduced and accumulated in iPSCs from their parental cells [150].

The potential for tumorigenicity due to genetic differences in iPSCs is one of the highest safety concerns. Another issue with iPSCs’ genetic and epigenetic heterogeneity is that it may impair the differentiation potential of the cells and lead to unexpected phenotypes [151]. Further research initiatives, such as creating a specialized mutation database for iPSCs and establishing a standardized set of criteria to screen genetic variations, are imperative for assessing genomic stability. This evaluation is crucial as genetic variants within iPSCs could have significant functional and safety implications.

## 6. Conclusions and Future Perspectives

The liver is an important organ in the human body, and its metabolic, immune, digestive, homeostatic, and detoxification roles depend on efficient crosstalk and hierarchical organization. Organoid technology has received tremendous interest in liver bioengineering to mimic liver-like features and functions for biomedical investigations. Recent advances in organoid research indicate that this technology can be applied to culture and fabricate liver organoids for disease models, cancer research, viral infection models, and translational research [49,50,51,150]. Organoid-based approaches offer alternative mini-liver-like platforms for investigating cancer initiation, development, progression, and therapy response assessment. However, reported organoid-based models mimic only a fraction of the physiological activities and structural patterns of the liver. Therefore, a more classical multicellular-based organoid model system that recapitulates most, if not all, of the cellular architecture and complex signaling pathways of the liver is a supreme requirement for tissue bioengineering and the regenerative medicine field. To design and produce such hierarchical organoid models with proper distribution and patterning of the cells, a better understanding of the biomolecular composition of the liver scaffold matrix and its role in cell-specific support is urgently needed. The development of advanced biomaterials with cell-specific supporting roles of the liver scaffolding matrix might provide the desired physiochemical features and mechanical integrity. Other critical challenges that need to be resolved in order to realize organoid technology for wider application include a proper selection of biomaterial sources, appropriate cell types (parenchymal and nonparenchymal cells), vasculature embedding, and quality control of the cultures. Besides the biomaterials and cells, technological obstacles also need to be addressed.

## Figures and Tables

**Figure 2 bioengineering-11-00346-f002:**
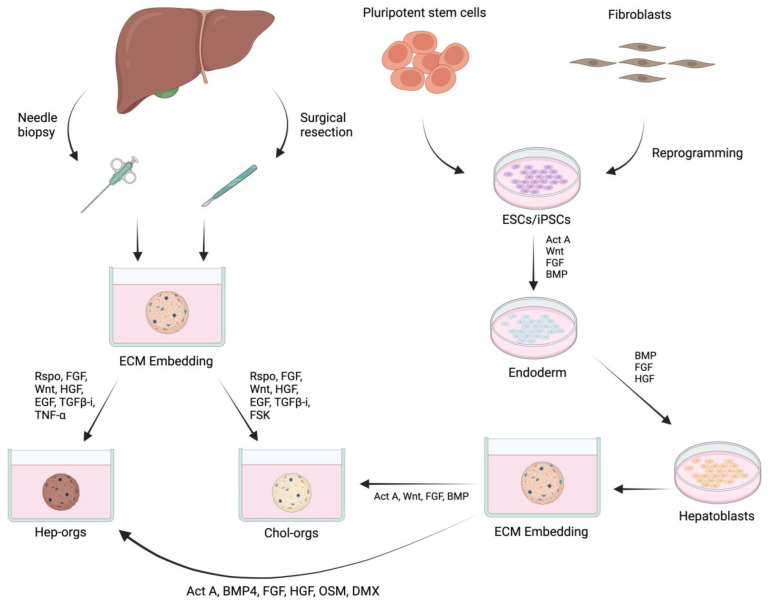
Conceptual diagram of the process of preparing cells from various sources to produce liver organoids. Abbreviations: Rspo, R-Spondin; TGFb-I, transforming growth factor beta inhibitor; TNFα, tumor necrosis factor alpha; ESCs, embryonic stem cells; FGF, fibroblast growth factor; FSK, forskolin; HGF, hepatocyte growth factor; iPSCs, induced pluripotent stem cells; Act A, activin A; BMP4, bone morphogenetic protein 4; DMX, dexamethasone; ECM, extracellular matrix; EGF, epidermal growth factor; OSM, oncostatin M; RA, retinoic acid.

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
