# Peer review of "Bioengineered Organoids Offer New Possibilities for Liver Cancer Studies: A Review of Key Milestones and Challenges"

_bioengineering, 2024, doi:10.3390/bioengineering11040346_

Round 1

Reviewer 1 Report

Comments and Suggestions for Authors

 The review article Bioengineered organoids offer new possibilities for liver cancer studies: A review of key milestones and challenges focuses on the establishment and application of liver organoid models in vitro for liver cancer research. Authors discuss strategies used to construct organoid models, current limitations and future prospects.  I recommended this work to be accepted after some corrections:

 Line 114 – cocorkers  - need to be checked

Line 198  self-self-assembly  - need to be checked

Line 448 – 452 repeated sentence- need to be checked

Line 465 repeated phrase- need to be checked

Author Response

We appreciate the reviewers' views on the paper.

Comment 1: Line 114 – cocorkers  - need to be checked

Response 1:  Revised

Comment 2: Line 198  self-self-assembly  - need to be checked

Response 2: Revised

Comment 3: Line 448 – 452 repeated sentence- need to be checked

Response 3: Revised

Comment 4: Line 465 repeated phrase- need to be checked

Response 4: Revised

Reviewer 2 Report

Comments and Suggestions for Authors

The review article “Bioengineered organoids offer new possibilities for liver cancer studies: A review of key milestones and challenges” by Jabri et al. is an interesting topic given the current status of liver cancer which is expected to reach 1 million cases by 2025. The authors have attempted to touch on most of the relevant sub-topics such as current treatment strategies, background and introduction to organoids, bioengineering strategies for liver organoids, limitations of organoids etc. Although, the paper is nicely written and well-structured it is not able to grab reader’s attention and hook them to the article. The opportunity to provide interesting information on organoids and liver cancer has not been utilized to its full extent. In my opinion, there are some shortcomings which can be improved by complying with the following suggestions and comments.

Firstly, in section 1.1, the heading indicates “current methodologies of liver cancer treatment” whereas the body of the section explains all the imaging techniques (LI-RADS, EASL, CEUS,CECT,CE-MRI) used for diagnosing liver cancer and evaluating the treatment response which is not equivalent to the treatment of cancer. The current gold standards of liver cancer treatment are chemotherapy and radiation which is not considered in the section. If the authors want to stick to the text they should rethink the section heading. In my experience, it would be beneficial for the article if the authors are able to shed some light on the current methods of liver cancer treatment.

Secondly, in section 2, I was expecting to gain some insights on the first organoid model developed ever. As the section is titled “a brief history of organoids” it should be presenting facts about all the organoids not just liver organoids. A timeline or a table about the development of liver organoids would be very fascinating from a reader’s point of view.

Thirdly, in section 4, where authors talk about bioengineering strategies for growing liver organoids, why did they chose to discuss only microfabrication strategies and excluded other approaches? If they want to provide perspective on strategies and not a single strategy I would suggest they add a short paragraph on 3D bioprinting of liver organoids which is one of the many hot topics in tissue engineering these days. I noticed that authors have cited references from bioprinting papers and journals, but never mentioned anything about it in the article. I am sure they would have a good reason not to. In my opinion, including 3D bioprinting as one of the strategies would significantly improve the reach and interest of the readers.

Minor comments:

1.     There are a lot of spelling and grammar mistakes. For example,

·      Page 4, line 183 “occours”

·      Page 4, line 187 “medicine”

·      Page 4, line 198 “self-self-assembly”

·      Page 10, line 367 “Oroyxlin”

·      Page 10, line 381, figure 3 caption “imaing”

·      Page 13, line 495 “onne”

·      Page 13, line 521 “to to”

·      Page 12, line 489 “concerns concerning”

·      Page 12, line 485 “to save experimental costs” is mentioned twice in the same sentence.

A thorough check on spelling and grammar should be performed.

2.     What does “CCC” denote in figure 2?

3.     Page 10, line 374, what are HCC PDL lines?

4.     What do all the acronyms in figure 3 stand for? Nothing is explained except ICC.

Comments on the Quality of English Language

There are a lot of spelling/typo errors which should be throughly checked. A few grammatical mistakes are also identified which should be fixed before publication.

Author Response

Comment 1: Firstly, in section 1.1, the heading indicates “current methodologies of liver cancer treatment” whereas the body of the section explains all the imaging techniques (LI-RADS, EASL, CEUS,CECT,CE-MRI) used for diagnosing liver cancer and evaluating the treatment response which is not equivalent to the treatment of cancer. The current gold standards of liver cancer treatment are chemotherapy and radiation which is not considered in the section. If the authors want to stick to the text they should rethink the section heading. In my experience, it would be beneficial for the article if the authors are able to shed some light on the current methods of liver cancer treatment.

Response 1:  Thank you for your valuable suggestion. As suggested, we revised the section and included additional contents/cited the relevant references (newly added text and references are highlighted in yellow).

Comment 2: Secondly, in section 2, I was expecting to gain some insights on the first organoid model developed ever. As the section is titled “a brief history of organoids” it should be presenting facts about all the organoids not just liver organoids. A timeline or a table about the development of liver organoids would be very fascinating from a reader’s point of view.

Response 2: Added the timeline graph and cited the relevant references (newly added references are highlighted in yellow).

Comment 3: Thirdly, in section 4, where authors talk about bioengineering strategies for growing liver organoids, why did they chose to discuss only microfabrication strategies and excluded other approaches? If they want to provide perspective on strategies and not a single strategy I would suggest they add a short paragraph on 3D bioprinting of liver organoids which is one of the many hot topics in tissue engineering these days. I noticed that authors have cited references from bioprinting papers and journals, but never mentioned anything about it in the article. I am sure they would have a good reason not to. In my opinion, including 3D bioprinting as one of the strategies would significantly improve the reach and interest of the readers.

Response 4: We thank the reviewer for his suggestions. This review primarily focuses on liver cancer organoids. We omitted bioprinting topic for two reasons. (1) the bioprinting of liver organoids field is still in its infancy, and (ii) We aim to submit a separate updated review on bioprinting technology for tissue and organoids models by the end of April.

 Comment 4:

Minor comments:

  1. There are a lot of spelling and grammar mistakes. For example,
  • Page 4, line 183 “occours”

Response : Revised

  • Page 4, line 187 “medicine”

Response : Revised

  • Page 4, line 198 “self-self-assembly”

Response : Revised

  • Page 10, line 367 “Oroyxlin”

Response : Revised

  • Page 10, line 381, figure 3 caption “imaing”

Response : Revised

  • Page 13, line 495 “onne”

Response : Revised

  • Page 13, line 521 “to to”

Response : Revised

  • Page 12, line 489 “concerns concerning”

Response : Revised

  • Page 12, line 485 “to save experimental costs” is mentioned twice in the same sentence.

Response : Revised

A thorough check on spelling and grammar should be performed.

  1. What does “CCC” denote in figure 2?

Response : Revised

  1. Page 10, line 374, what are HCC PDL lines?

Response : Revised

  1. What do all the acronyms in figure 3 stand for? Nothing is explained except ICC.

Response : Revised

Comment 4: Comments on the Quality of English Language There are a lot of spelling/typo errors which should be throughly checked. A few grammatical mistakes are also identified which should be fixed before publication.

Response 4:

 Thank you so much for your comments; we appreciate all your time. We have reviewed the manuscript and fixed these issues throughout. The revised manuscript was checked by a native English speaker, who made the necessary amendments.

Round 2

Reviewer 2 Report

Comments and Suggestions for Authors

Accept in present form